# Development and Validation of the Mexican Public Open Spaces Tool (MexPOS)

**DOI:** 10.3390/ijerph19138198

**Published:** 2022-07-05

**Authors:** Catalina Medina, Annel Hernández, Maria E. Hermosillo-Gallardo, Célida I. Gómez Gámez, Eugen Resendiz, Maricruz Morales, Claudia Nieto, Mildred Moreno, Simón Barquera

**Affiliations:** 1Centro de Investigación en Nutrición y Salud, Instituto Nacional de Salud Pública, Cuernavaca 62100, Mexico; annel_chivas@hotmail.com (A.H.); maria.hermosillo.gallardo@gmail.com (M.E.H.-G.); maricruzmoraleszarate@gmail.com (M.M.); claudia.nieto@insp.mx (C.N.); sbarquera@insp.mx (S.B.); 2Departamento de Arquitectura, Urbanismo e Ingeniería Civil, Universidad Iberoamericana, Mexico City 01219, Mexico; celida.gomez@ibero.mx; 3Prevention Research Center, Brown School, Washington University in St. Louis, St. Louis, MO 63130, USA; eugen@wustl.edu; 4Escuela de Ingeniería y Arquitectura (ESIA), Instituto Politécnico Nacional (IPN), Mexico City 07340, Mexico; milamovi@hotmail.com

**Keywords:** public open spaces, features, validity, tool, Mexico

## Abstract

Public open spaces (POS) are “publicly owned spaces such as parks, green areas, squares, marketplaces, streets and highways which are of public access”. Some attributes could increase or decrease participants’ attendance. Thus, reliable and valid audit tools are needed in order to measure these attributes. This study aimed to develop and validate a tool to assess POS features within the Mexico City context. The Mexican Public Open Spaces Tool (MexPOS) was developed based on (1) two validated POS audit tools, (2) several visits to the POS, (3) pilot testing, and (4) multiple work sessions with a group of specialists. The original tool included 181 items divided into nine sections. Trained personnel visited and evaluated 944 POS in Mexico City. An exploratory factor analysis was performed to examine the construct validity of the items and the relationship between the subscales. The final model resulted in seven factors: (1) Food and Wellness Environment (α = 0.15), (2) Maintenance (α = 0.81), (3) Amenities (α = 0.72), (4) Legibility (α = 0.59), (5) Security (α = 0.48), (6) Perceived Environment (α = 0.65), and (7) Urban Environment (α = 0.58). Our study highlights the relevance of using a validated tool to measure POS characteristics related to participants’ attendance to help assess infrastructure improvements and identify priority areas for changing socio-urban environments for physical activity.

## 1. Introduction

Public open spaces (POS) are “publicly owned spaces such as parks, green areas, squares, marketplaces, streets and highways which are of public access” [1]. According to several studies, POS provide opportunities for physical activity [2,3,4], promote mental and physical health [5], personal well-being, and socialization [4], reduce stress [6] and mortality [7], grant direct or indirect economic benefits [6], and increase sedentary behaviors across different population groups [8], as well as, in some contexts, food consumption [9]. However, the proximity [10] and the condition of park features could be relevant to visiting these places [10,11].

Some studies have evaluated POS globally. For instance, the project Public Spaces is an American organization, with an international impact that aims to identify the characteristics of successful POS. This project evaluated thousands of POS in the world and concluded that a “successful” space had four key qualities: (1) accessibility (i.e., unrestricted access), (2) availability of a range of activities, (3) convenient and pleasant space, and (4) adequate space to meet and socialize [12]. Similarly, Jan Gehl developed criteria to evaluate the qualities of POS in cities and divided the criteria into three groups: (1) protection against vehicular traffic, accidents, crime, and unpleasant sensory experiences (e.g., smell of litter), (2) opportunities to walk, stand, sit, play and exercise, and (3) opportunities to enjoy the POS, its aesthetic quality, and positive sensorial experience [13]. However, this could change depending on age group [2] and country income [14].

A systematic review of qualitative and quantitative studies on the characteristics of POS that influence adolescents’ physical activity found that the presence of trails, walking paths, playgrounds, and sports fields are associated with physical activity among this population [2]. In the case of adults and children, the good maintenance and cleanliness of the POS are important, as well as the availability of toilets, drinking fountains, benches, shelters [14], illumination, exercise equipment, and playgrounds [15]. As to safety-related items, the presence of homeless people, broken glass, trash, feces, lack of maintenance, and graffiti discouraged park use among the population [14,15]. Sensory characteristics such as pleasant smells, the sensation of fresh air, and the sounds of nature were also recurring characteristics in the literature that are related to the use of POS and physical activity [2,14]. Lastly, a study in 163 children (aged 8–9) found an increase of 25 min of moderate to vigorous physical activity per day if there was a playground in the POS [16]. In agreement with these attributes, various tools have been developed to assess objectively the physical aspects of the POS globally [17,18,19,20,21,22].

The Bedimo-Rung Assessment Tool–Direct Observation (BRAT-DO) is an extensive instrument, mainly used by researchers and health professionals, that evaluates the visible characteristics of the park, it has 181 items distributed into different sections such as generalities of the park, aesthetics, concession stands, drinking fountains, sports fields, among others [17]. Likewise, the Environmental Assessment of Public Recreation Spaces Tool (EAPRS) evaluates the POS according to their potential functionality [18]; for example, when assessing a children’s playground, EAPRS not only evaluates the functionality of this facility for children but how it could be used by adolescents and adults as well. It is the most complete tool for POS assessment (752 items), and similar to BRAT-DO, its use is recommended for researchers and health professionals. On the other hand, other audit instruments such as the Community Park Audit Tool (CPAT) [19], the Physical Activity Resource Assessment (PARA) [20], and the Quality of Public Open Space Tool (POST) [21] are briefer (25–50 items) than BRAT-DO and EAPRS, they are more user friendly (e.g., definitions and instructions are built into the tool), and in the case of POST, include diagrams of fields and spaces to map the arrangement of paths, trees and other elements inside the POS. However, the abovementioned audit tools were developed to be used in high-income countries (HICs).

Even though the tools mentioned above assess the physical attributes of POS, they fail to assess in depth some important aspects of POS in Mexico, such as the built environment around the POS (e.g., streets around the POS, streets with pedestrian traffic lights, and safe crossings around POS) and perception of safety (e.g., police presence, security cameras). A population-based study on adults in Mexico (*n* = 629, aged 20–65) found that easy access and proximity to parks, aesthetics, and safety from crime were associated with moderate to vigorous physical activity; moreover, the perception of park safety was found to be a moderator of the association between physical activity and park use [23].

Currently, there is no validated tool for measuring POS attributes in Mexico. Having a validated and objective tool could help administrators, city planners, urbanists and stakeholders to create and/or modify some of the POS’s features that can reduce attendance and increase healthy lifestyles. Therefore, this study aimed to develop and validate a tool that could assess POS features within the Mexico City context.

## 2. Materials and Methods

### 2.1. Development of the Audit Tool and Pilot Testing

A first meeting was held at the National Institute of Public Health Mexico (INSP) with a group of specialists in physical activity (*n* = 2), nutrition (*n* = 6), urbanism (*n* = 2), architecture (*n* = 2), and public policy (*n* = 1) at the beginning of 2016. During this session, some characteristics of POS that could help foster active lifestyles were listed and discussed. Among these characteristics, they highlighted the presence of trees, pollution, safety, accessibility, location, lighting, presence of pets, advertising of foods that promote a sedentary lifestyle, sale of energy-dense foods, and hygiene.

Three emblematic parks in the center (El Sope, Chapultepec) and south (Viveros de Coyoacan and Bosque de Tlalpan) of Mexico City were visited. This was a convenient selection of parks focused on spaces that have high participation of people who perform physical activity. These visits helped us to verify some of the characteristics listed at the first meeting and to include some others found inside (e.g., trails, lighting, amenities) and outside (e.g., zebra crossing, advertising) of the POS. After this visit, 16 items were created, including those related to the presence of pets, availability, quantity, quality, and cost of bathrooms, availability of garbage containers, smoke advertising, food, and beverage advertising, availability of drinking fountains, availability of energy-dense foods around the POS, availability of police stations, public transportation stops, availability of lighting, parking lots, outdoor gym and management of the POS. Later, this list was compared and complemented with the characteristics assessed by the Bedimo-Rung Assessment Tool–Direct Observation (BRAT-DO) [17] and the Environmental Assessment of Public Recreation Spaces Tool (EAPRS) [18]; both validated instruments used to evaluate park characteristics in the US. Twenty items from BRAT-DO regarding activity areas, hours of operation, advertisements, landscape condition, hygiene, the attractiveness of the park, drinking fountains, workers within the park, streets, and sidewalks’ characteristics around the park were included [17]. In addition, twelve items related to characteristics of walking and running tracks and resting areas were added from EAPRS, as this tool measures all the elements of the parks separately [18]. Items and answers from both instruments were adapted to the Mexican context.

In the following meetings at the INSP, urbanists included some other questions related to mapping, quality of lighting, rules of procedure, name of the POS, areas of the park open to the public, memberships, accessibility, public transportation availability, streets around the POS, and security elements. Physical activity experts included items related to drinking fountains and their use. Nutritionists redesigned questions about food and beverage advertising and availability.

Next, the new version of the audit tool was pilot tested by a group of urban planners in a metropolitan park in west-north Mexico City (Parque Bicentenario). During this visit, several adaptations were made to: (1) reduce the number of items, (2) eliminate items that measure similar characteristics or those that were not context-specific, (3) improve the wording of the items, and (4) include other POS attributes. Subsequently, two external volunteers from the research group applied the tool and provided feedback on the clarity of the tool’s instructions. The original version of the tool had 181 items and the following sections: (1) General park information, (2) General park features (e.g., activities within the POS, information and signaling, aesthetics), (3) Accessibility, (4) Environment around the POS, (5) Roads/internal routes/tracks within the POS, (6) Security, safety and lighting, (7) Facilities and amenities (e.g., toilets, drinking fountains, litter bins, benches), (8) Health, nutrition, and hygiene (e.g., medical services inside the park, food advertising, food establishments, hygiene-related to pets), and (9) Maintenance (e.g., agency in charge of the POS). Items were rated with three types of response scales: (1) Likert scales (three and five points ranging from totally agree to totally disagree), (2) binary scales (yes/no), and (3) ordinal scales (e.g., <10, 10–20, >10). The original version of the questionnaire in Spanish can be found in Appendix A.

### 2.2. Selection of Public Open Spaces

For this article, a POS was defined as green spaces (e.g., parks), grey areas (e.g., plazas), or natural environments (e.g., ecological reserves with access to people, regardless of their size, with recreational purposes, accessible to the general public and mostly free of charge or with an entry fee less than five US dollars. The selection of POS was made through a review of different public databases and a creation of a list compiling all of Mexico City’s POS. First, national public geostatistical databases using the digital map platform from the National Institute of Statistics and Geography (by its acronym in Spanish: INEGI) [24] were reviewed and protected natural areas, green areas, parks, roundabouts, ridges, and other green areas were visually identified and selected on the digital map. The reviewed databases included the Protected Natural Areas from the National Commission of Natural Protected Areas (by its acronym in Spanish: CONANP), the National Geostatistical Framework 2017: Green Areas/Parks/Roundabouts/Ridges, and the Land Use and Vegetation 2011 database. Then, the Ministry of Environment (by its acronym in Spanish: SEDEMA) and the Ministry of Environment and Natural Resources (by its acronym in Spanish: SEMARNAT) databases were reviewed to identify the parks and green areas managed by these institutions. Finally, the land use Geographic Information System of the Ministry of Urban Development and Housing (by its acronym in Spanish: SEDUVI) website was reviewed [25]. This website was used to verify the location and land use of each POS in Mexico City. Previous searches were complemented through a visual review of the Roji Guide (i.e., printed map of Mexico City) and physical scanning during fieldwork.

All POS that fitted our definition were included in the final sample. POS that were difficult to access (e.g., fenced), required an annual membership (e.g., private sports clubs), were private (e.g., spaces in residential areas) or did not serve a recreational purpose (e.g., agricultural-related fields) were excluded from the final sample.

A total of 944 public open spaces (POS) distributed in all the municipalities of Mexico City were assessed between August 2017 to February 2018, with the original version of the tool by trained personnel. A list of POS was distributed among 3 fieldworkers at convenience. The most remote and insecure spaces (based on interviewers’ perceptions) were evaluated by the entire team (3 fieldworkers). On average, each fieldworker visited 15 POS each week. POS were categorized based on their typology into: (1) metropolitan park: ≥10,000 m^2^, (2) local park: 3000 to 10,000 m^2^, (3) neighborhood park: 400 to 3000 m^2^, (4) pocket park: 100 to 400 m^2^, (5) roundabout: has a higher proportion of sealed soil, vegetation or planters, and is planned to regulate vehicular traffic, (6) boulevard (Alameda): public space of up to 80,000 m^2^, with vegetation and bare or covered soil and paths or corridors for pedestrian traffic, (7) square (plaza): public space of up to 5000 m^2^, which has a greater proportion of sealed soil, has arboreal and/or shrubby vegetation. Planned for recreation, rest, or relaxation, (8) remnant: linear green space generally enabled along disused railway lines, rivers, streams, canals, and urban voids, and (9) garden: public space of up to 5000 m^2^, which has a higher proportion of sealed soil, has arboreal and/or shrubby vegetation, and is planned for recreation, rest, or relaxation [26].

### 2.3. Statistical Analysis

An exploratory factor analysis (EFA) was performed to examine the validity of the items and the relationship between the subscales. An EFA is a multivariate technique used to uncover the underlying structure of a set of observed variables (i.e., indicators) and the construct they measure (i.e., factor/latent variable) [27]. In the case of our tool, whether its items (i.e., indicators) have similar patterns of responses and therefore can be grouped and create a subscale/section (i.e., factor/latent variable).

Prior to the EFA, several assumptions (type of variable, linear associations, sample size, and absence of outliers) and pre-tests were checked to corroborate that this analysis could be performed. Regarding the assumptions, all data that were put into the model were either continuous or ordinal (e.g., Likert scales), and Yes/No variables (binary data) were treated as ordinal data assuming an underlying order “yes is higher than no” as has been done in previous research [28]. The rest of the nominal data (e.g., open answers) in the questionnaire were not part of the analysis, as they did not provide any quantitative value. Scatter plots were performed among the variables to check the linear associations and box plots to test for outliers. As for the sample size, factor analysis requires a sample size bigger than 500 (although recommendations vary) and a subject-to-variable ratio higher than 2:1; in the case of the data of the study, the sample size was 944 with 243 variables with a ratio of ≈4:1.

Concerning the pre-tests, the Kaiser–Meyer–Olkin Measure of Sampling Adequacy (KMO) was performed to reflect the sum of partial correlations relative to the sum of correlations. KMO results vary between zero and one, where a value closer to one is better and a value lower than 0.5 indicates that an EFA might not be appropriate [29]. The KMO result of our data was 0.75, indicating the suitability of an EFA. In addition, Barlett’s Test of Sphericity was performed to test the hypothesis that the correlation matrix is an identity matrix (i.e., square matrix with ones on the main diagonal and zeros elsewhere), meaning there would not be correlations between the variables; hence, the test needs to be statistically significant (i.e., *p* < 0.05) to reject the hypothesis [29]. In the case of this study, Barlett’s Test of Sphericity was *p* = 0.001.

Once the assumptions and pre-tests had been carried out the EFA was performed using principal factor analysis. To determine the number of factors, eigenvalues were calculated using the “*factor*” command in STATA. Eigenvalues are indicators of the variance explained by a factor, they should be greater than one in order to account for at least as much variance as a single variable, hence the factors with an eigenvalue greater than one were extracted resulting in 13 factors. Thereon, commands “*rotate*” and “*sortl*” were used to rotate the factors for easier interpretability and to arrange the variables in descending order into the 13 factors each are loading. Alphas were calculated for each of the factors. All analyses were performed in STATA (Version 13), College Station, TX.

## 3. Results

### 3.1. POS Distribution

A total of 944 POS were mapped and evaluated. Based on their typology, 26.7% were neighborhood, 24.5% locals, 20.9% metropolitans, 10% remnant, 7.3% squares, 4.5% pocket, 4.4% gardens, 1% roundabout, and 0.7% boulevards. Based on the municipality, the highest concentration of POS was located in four municipalities: Cuauhtémoc, Coyoacán, Iztapalapa, and Gustavo A. Madero (52.8%). In total, 14.4% of the POS was found in Cuauhtémoc municipality. Cuajimalpa, Magdalena Contreras, Milpa Alta, Xochimilco, and Iztacalco municipalities have least number of POS. Specifically, Milpa Alta, the second-largest municipality in Mexico City, possesses the lowest concentration of POS (0.3%) (Figure 1).

### 3.2. Exploratory Factor Analysis

From the first results of factor loadings (Model I) (Appendix A), a group of researchers gathered to discuss the item loadings coherence. According to researchers’ expertise and knowledge, some items were moved to factors in which they theoretically fit better; this is a standard practice in EFA analysis [29]. As a result, items in factors 7, 10, 11, 12, and 13 were allocated to other factors. Factors with less than 0.3 were suppressed [30], and scores greater than 0.4 were considered to be stable [31]. Name titles were designated to each factor and alphas were recalculated resulting in Model II (Appendix A). This process was repeated, and Factor 2 “Amenities and Maintenance” (α = 0.71) was divided into the factors “Amenities” and “Maintenance”, as there were two clear latent variables mixed, according to the group of researchers; consequently, alphas for these two factors increased to 0.72 and 0.79, respectively (Model III) (Appendix A). Finally, one last revision was made, in which factors 6 and 9 were united to explore for a better fit, resulting in the final model (Model IV).

The final model (Model IV) is presented in Appendix A, and included seven factors: (1) Food and Wellness Environment (α = 0.15), (2) Maintenance (α = 0.81), (3) Amenities (α = 0.72), (4) Legibility (α = 0.59), (5) Security (α = 0.48), (6) Perceived Environment (α = 0.65), and (7) Urban Environment (α = 0.58). These factors conformed to the Mexican Public Open Spaces Tool (MexPOS). The entire process is described in Appendix A.

In the Food and Wellness Environment factor, items (*n* = 6) were related to the land use around the park and the food and beverage advertising inside and outside the POS. All items were reverse items except for the land use around the POS, which was mainly institutional. In the Maintenance factor, items (*n* = 21) were related to the condition of the facilities (e.g., walking paths, seating areas, adequate landscape) inside the park. Reverse items in this factor were the presence of garbage around the POS, hazardous waste (e.g., alcohol containers, needles), and animal feces.

The Amenities factor comprised items (*n* = 44) related to the courts (e.g., fronton, football), paths (e.g., jogging, bike), outdoor gym equipment (e.g., rowing machine, hoops, elliptical trainer), and other POS features (e.g., appropriate lighting, botanical gardens, ponds) in the POS. Moreover, in the Legibility factor, items (*n* = 15) were relevant to the references or signs indicating characteristics of the POS (e.g., name, appropriation, schedule of operation), activities in the POS (e.g., historical/educational/artistic features, events), and rules inside the POS (e.g., dogs must be kept on a leash, no parking, recyclable bins). The items (*n* = 11) in the Security factor were features related to the protection against deliberate threats inside the POS (e.g., police stations, security cameras, panic buttons) and to elements that might make a person feel unsafe (e.g., poor lighting, previous safety incidents, presence of threatening people). It should be mentioned that four out of eleven items (6.8 to 6.11) did not reverse load as expected.

The Perceived Environment factor comprised items (*n* = 14) relevant to the sensory aspects of the POS (e.g., bird sounds, pleasant smells, general attractiveness), as well as the number of trees and shade available. Reverse items were related to car traffic sounds, excessive noise, and people smoking inside the POS. Furthermore, in the Urban Environment factor, items (*n* = 17) were related to the type of streets around the POS (e.g., primary, secondary, tertiary), the presence of signaling devices such as pedestrian crossing lights and motorized vehicles stoplights, and ramps to guarantee access to the POS. Reverse items were having most streets around the POS primary, having sidewalks in a “regular” state, and having public transport stops and stations near the POS.

## 4. Discussion

The purposes of this study were to develop and validate a tool that could assess POS features within the Mexico City context. A team of experts, who integrated evidence and developed processes, designed and validated the MexPOS tool. The original version of the tool included 181 items divided into nine sections (general park information, general park features, accessibility, the environment around the park, roads/internal routes/tracks within the park, security, safety and lighting, facilities and amenities, health, nutrition and hygiene, and maintenance). However, after conducting the factor analysis, the final model comprised seven factors or sections including food and wellness environment, maintenance, amenities, legibility, security, perceived environment, and urban environment. Internal consistency estimates for the latent variables showed strong alpha coefficients for Maintenance (α = 0.81), Amenities (α = 0.72) and Perceived Environment (α = 0.65), acceptable estimates for Legibility (α = 0.59), and Urban Environment (α = 0.58), and unacceptable values for Food and Wellness Environment (α = 0.15) and Security (α = 0.48) [32].

### 4.1. POS Definition

At the time of this research, there was not a unique definition of POS, each public organization or government used a different definition, making it difficult to have a clear definition or understanding of these spaces. POS were also classified in accordance with the local urban norms defined by the federal, state, and local governments, where these spaces are defined mainly as green spaces and grey areas. However, a definition by the Authority of Public Spaces [26], which classify these areas by their dimensions and uses, was considered, and used as follows: neighbourhood, locals, metropolitans, remnant, squares, pocket, gardens, roundabout, and boulevards.

### 4.2. Development of the MexPOS

Although there are many available instruments to measure POS worldwide [17,18,19,20,21,22], many of them lack characteristics related to the place to be evaluated. For example, the EAPRS [18] and the BRAT-DO [17] are questionnaires used mainly in HICs, whose main characteristics are based on measuring aspects not related or not relevant to the Mexican context, such as water-related amenities, drinking fountains, and good maintenance. This means that the adaptation of other instruments to different contexts can be somewhat complicated, specifically in low-middle income countries (LMICs) whereas security, presence, and/or maintenance of amenities, hygiene, health (e.g., spit, littering, taking drugs, feces, trash) and availability and/or advertising of healthy/unhealthy products could be more prevalent [33,34]. Thus, caution must be taken when using a tool that does not consider local characteristics, because this could potentially affect the overall assessment of the POS. For such reasons, aspects such as food advertisement in the park, security elements (panic buttons, security cameras), people’s behaviors (people smoking) and medical services, were included in the MexPOS to provide an insight into the acute characteristics of these spaces.

Second, there are some POS characteristics that could differ between countries and cities. This is the case of a big-sized and urbanized city such as Mexico City. This city possesses more than 900 POS divided into nine typologies. Despite considering a variety of characteristics within the instrument, some could be missed or not considered. For instance, the availability of restrooms could be a mandatory amenity within metropolitan parks, but not for the pocket, remnant, or plaza areas. This could generate an unfair estimation of the POS’ features, especially if the instrument was designed to generate an overall score (e.g., EAPRS) [18]. In this case, MexPOS was mainly designed to describe the internal and external characteristics of the POS. This can help policymakers in evaluating the impact of implemented infrastructure improvements (e.g., playgrounds, running tracks, lighting) and in identifying priority areas for new investments.

### 4.3. Mapping

Park identification using national databases could be challenging in some countries. This is the case for Mexico City, where more than three public databases were searched and several physical scans were performed to generate a list of POS. In contrast, HICs have databases open to the public that describe park characteristics in detail [35,36,37]. Therefore, if governments intend to increase the number of users within the POS, it is imperative to have a universal database stratified by region, city, and country, which could increase the motivation and willingness to visit and use POS.

### 4.4. Similarities with Other Studies

Based on validity values, it is complex to compare the obtained internal consistency estimates with other POS audit tools, as they do not report internal consistency measures. CPAT, EAPRS, PARA, and POST only report inter-rater reliability estimates (κ = 0.60–0.77) [17,18,19,20,38], and BRAT-DO reports inter-rater reliability (κ = 0.87) and domain validity (κ = 0.79) by comparing the results obtained by fieldworkers (physical assessment of the POS) with a gold standard (i.e., expert opinion) [17]. A validated audit tool to objectively measure the characteristics of parks, grey areas, and natural environments is key for assessing their relationship with physical activity.

### 4.5. Factor Analysis of the MexPOS

The Cronbach alphas for Security and Food and Wellness Environment factors are low. As to Security, there were four out of eleven items that were expected to reverse load because of their nature. Items (6.8) Is there any evidence of threatening people or behavior? (6.9) Is there vandalism inside the park? (6.10) Is there poor street lighting in the area around the park? and (6.11) Have there been any security incidents so far this year? are measured as if the presence of it (i.e., “Yes”) adds up positively to the construct of Security. A possible reason for this is the complexity and “hour dependency” of measuring these items, for example, items (6.8) and (6.10) would depend on the time of the day of the fieldworker’s visit and their ability to record security incidents; (6.11) might depend on whether there is a park administrator to ask to or if incidents have been reported by the media.

Food and Wellness Environment has also a low alpha factor (0.15). The measurement of the food environment is becoming relevant given its contribution to the health status of the population [9]. Specifically, in recent years, an increase in the availability and marketing of high-energy-dense foods was observed within POS [39,40]. This has aroused a certain interest in the authorities to measure and regulate these products. Although some items related to the food environment were proposed and carefully included by a group of nutrition experts, the original audit tool failed to include the key items that measure this construct. Caution should be taken when measuring the food and wellness environment using this instrument. Thus, future studies would have to consider some other variables that could strengthen this component.

Regarding the Maintenance and Amenities factors, it could be argued that the high Cronbach alphas might be due to the high number of items in the analysis in both latent variables; nevertheless, these values are not so high (>0.95) as to indicate redundancy of the scale items [32]. As for items from the Legibility and Urban Environment factors, their internal consistency is slightly above the threshold limit (>0.50) for being acceptable; hence, it could be stated that their items are sufficient for measuring both constructs.

## 5. Conclusions

### Strengths and Limitations

Among the strengths of the study is the use of a large dataset generated with complete information for 944 POS. To our acknowledge, this is the most comprehensive POS database created within Mexico City. The development of a Mexican context questionnaire that considered various elements of the context that contributes to healthy lifestyles (e.g., facilities for physical activity, food availability, food advertising, water availability, medical services) can be seen as a strength. Additionally, the corroboration of the assumptions and pre-tests contribute to a clear basis for using an EFA to assess construct validity. Moreover, the participation of a diverse group of field experts to define, based on the analysis results, the final EFA model adds to the strengths of the article. Finally, the MexPOS can be used in other cities of LMICs where the built environment can be similar to that of Mexico City.

This study also has some limitations that should be addressed. First, the study is only representative of a single city in the country, and so far, has not been tested in other less urbanized areas. Second, the number of POS could be higher or lower within current years. For instance, pocket parks are no longer in operation. Third, some of the perception items (such as insecurity, and the general appearance of the park) were difficult to capture, which could have altered the alphas.

In conclusion, The MexPOS is a valid tool to measure the constructs of maintenance, amenities, legibility, perceived environment, and urban environment in POS. Although future research is needed in order to estimate the test–retest reliability, inter-rater reliability, and validity of this instrument, the MexPOS could help policymakers in evaluating the pre and post impacts of redesigning or renovating POS. In addition, this instrument could be used to improve and identify priority areas for changing socio-urban environments for physical activity based on typologies and/or municipalities, and to promote POS with ideal health conditions. This study could also contribute to making recommendations aimed to complement local and federal norms (e.g., those related to vulnerable groups, and insecurity).

## Figures and Tables

**Figure 1 ijerph-19-08198-f001:**
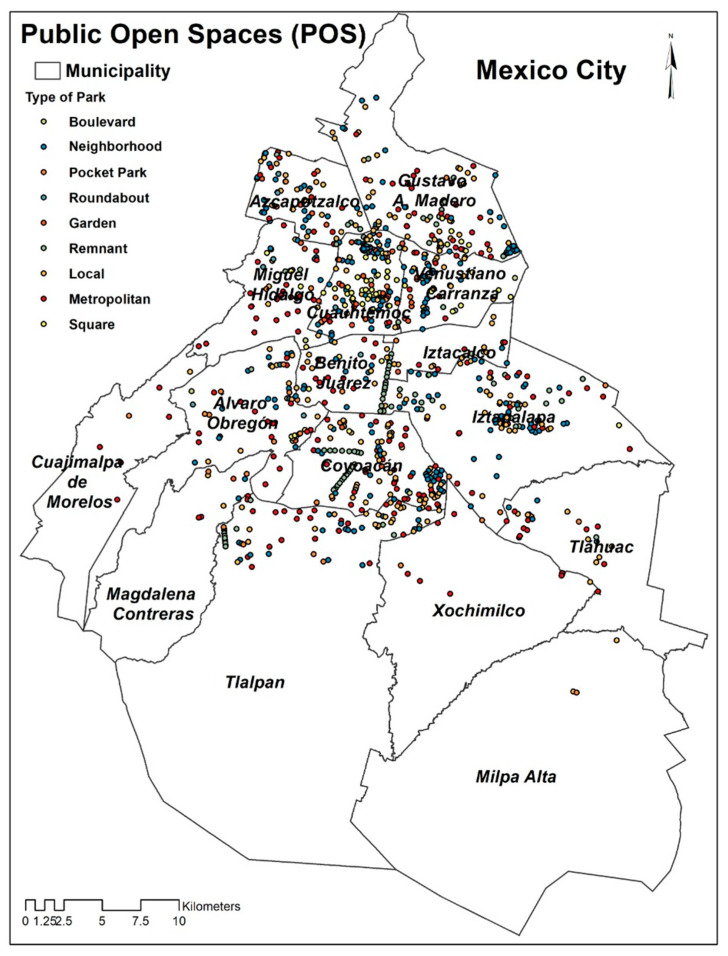
Distribution of POS according to municipality and typology.

## Data Availability

The data presented in this study are available on request from the corresponding author.

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
