# Peer review of "Development and Validation of the Mexican Public Open Spaces Tool (MexPOS)"

_ijerph, 2022, doi:10.3390/ijerph19138198_

Round 1

Reviewer 1 Report

Dear Authors,

The paper is interesting, but it should be done some improvments. First it is important to formulated hypothesis. After that it will be easer to write discussion. In my opinion results part is too short. It should be prepared with more details. Please write more in part conclusions, too. It will be great if You good add  some more actual publications connected with green areas in cities like this:

https://mdpi-res.com/d_attachment/sustainability/sustainability-13-01558/article_deploy/sustainability-13-01558-v2.pdf?version=1612345090

Author Response

Rev1

The paper is interesting, but it should be done some improvements. First, it is important to formulate a hypothesis. After that, it will be easier to write a discussion.

Thanks for the comment. We have added important information into the discussion section that we had omitted.

In my opinion, results part is too short. It should be prepared with more details. Please write more in part conclusions, too.

We have extended the results and discussion sections.

It will be great if you good add some more actual publications connected with green areas in cities like this:

https://mdpi-res.com/d_attachment/sustainability/sustainability-13-01558/article_deploy/sustainability-13-01558-v2.pdf?version=1612345090

Thanks for the recommendation. We have included and updated the references within the introduction section.

Reviewer 2 Report

The manuscript describes a study which adapts the developed POS tool for the use in Mexican cities. While the study may bear some practical implications, the research subject is not clear. POS is more than parks but this empirical study focussed on parks only. The applicability of the MexPOS to other types of POS is highly questionable.

Besides, I cannot see how the adapted scale was suitably validated. Externally, the authors should use other measurements (e.g. users' satisfaction or expert's subjective evaluation) to validate whether the measurements of the adapted scale really measure what they intend to measure.

Moreover, the authors did not justify why exploratory factor analysis (EFA) was used for validation. EFA is less powerful than confirmatory factor analysis (CFA).

There are quite many typos and grammatical errors in the manuscript. For example, it should be "last" rather than "las" in line 157. The authors should have the manuscript proofread by the professional English writer before submission.

Author Response

The manuscript describes a study that adapts the developed POS tool for use in Mexican cities. While the study may bear some practical implications, the research subject is not clear.

POS is more than parks but this empirical study focused on parks only. The applicability of the MexPOS to other types of POS is highly questionable.

The term POS (public open spaces) was used since there are some other open spaces, besides parks, that promote regular assistance and increase physical activity levels in our context. For example, roundabout “these open public spaces have a higher proportion of sealed soil, vegetation or planters. These are planned to regulate vehicular traffic. Further information on the definition of each POS is described within the “selection of public open spaces section” lines 183-194. In addition, a picture of this POS is included below:

Besides, I cannot see how the adapted scale was suitably validated. Externally, the authors should use other measurements (e.g. users' satisfaction or expert's subjective evaluation) to validate whether the measurements of the adapted scale measure what they intend to measure.

Thanks for the comment. For this study, we are performing “construct validity”. According to McGartland, et.al.,1 construct validity is the extent to which the test may be said to measure a theoretical construct or trait. In the second stage of the study, we will estimate inter-rater reliability, test-retest, and face validity.

Construct validity was described in the abstract section, statistical analysis, discussion, and conclusion sections.

Moreover, the authors did not justify why exploratory factor analysis (EFA) was used for validation. EFA is less powerful than confirmatory factor analysis (CFA).

This questionnaire was not based on a previously validated instrument, but rather it was a mixture of various elements including questions that other people considered relevant and that is why we used EFA instead of CFA.

There are quite many typos and grammatical errors in the manuscript. For example, it should be "last" rather than "las" in line 157. The authors should have the manuscript proofread by a professional English writer before submission.

The manuscript has been reviewed by an English-speaking professional before submission.

Reviewer 3 Report

This study develops and validates a tool to assess POS features within the Mexico City context. Methodologically, the Mexican Public Open Spaces Tool (MexPOS) was developed based on: 1) two validated POS audit tools, 2) pilot testing, and 3) multiple work sessions with a group of specialists. This is a decent study that the novelty is not really exceptional as many studies on POS quality assessment tools have been done. Each city or country has its own tool and most of them are well justified. One of the strong points highlighted in the study is that the tool has been tested on a large number of POSs, where 7 factors covering 243 items have been found.

Regardless of the above, there are a few suggestions and improvements that must be incorporated in the revised paper:

-In the abstract section, the bold word such as objective, methodology, etc. need to be reworded using proper sentences. For example, this paper or study's objective is to...

-Define HICs, and they are some studies that have been conducted in medium or upper medium income, e.g., see Ling et al., 2016 (https://www.thecommonsjournal.org/article/10.18352/ijc.618/#fn32), a study that has been carried out to assess a city's POS in Malaysia.

-elaborate line 81-82, what are the built environment attributes and safety features that are not sufficiently studied?

-For the problem or research gap/contribution of knowledge, it should be made clear. and the context or types of POS quality to be measured also needs to be clarified, e.g., urban parks or residential parks or both?

-Methodology is fine as the selected POSs are well justified. I am just thinking, the types and size of POS covered in the study are wide-ranging and diverse, how then this so called one size fits all assessment tool can be suitably and relevantly used? This needs to be discussed

-The statistical analysis is appropriate (EFA to validate the items under each factor) (unidimensionality analysis). However, factor loading cut off needs to be clarified...whether the authors used 0.5 or 0.6 or lower?

-whether the alpha figures here are representing the reliability of each factor? What analysis has been used to ensure its reliability?

For the results section, factor analysis and cronbach alpha have been adopted but i noticed there is one particular factor  with very low alpha value (less than 0.5, i.e., 0.15 for food and wellness environment), based on this, the items grouped under the factor are not consistency or reliable. I suggest removing them as this already indicated poor selections of items.

-Since the authors also mentioned the importance of kappa value (measuring inter-rater reliability), why did not the authors carry out this with the 3 evaluators for the parks? Inter rater reliability is one of the key components ensuring the reliability and hence validity of the tool.

-Discussion needs to specifically highlight the similarities and differences that this study has contributed to the existing literature. The current piece is not clear.

-Structure of sections is necessary that limitations and suggestions should be incorporated under the conclusion section.

-

Author Response

This study develops and validates a tool to assess POS features within the Mexico City context. Methodologically, the Mexican Public Open Spaces Tool (MexPOS) was developed based on: 1) two validated POS audit tools, 2) pilot testing, and 3) multiple work sessions with a group of specialists. This is a decent study the novelty is not really exceptional as many studies on POS quality assessment tools have been done. Each city or country has its own tool and most of them are well justified. One of the strong points highlighted in the study is that the tool has been tested on a large number of POSs, where 7 factors covering 243 items have been found.

Regardless of the above, there are a few suggestions and improvements that must be incorporated in the revised paper:

-In the abstract section, the bold word such as objective, methodology, etc. need to be reworded using proper sentences. For example, this paper or study's objective is to...

Thanks for the comment. We have removed bold words in the abstract.

-Define HICs, and they are some studies that have been conducted in medium or upper medium income, e.g., see Ling et al., 2016 (https://www.thecommonsjournal.org/article/10.18352/ijc.618/#fn32), a study that has been carried out to assess a city's POS in Malaysia.

We have deleted HICs from the sentences since we also included some information about other countries including Mexico. In addition, we do include the recommended reference in the introduction section.  

-elaborate line 81-82, what are the built environment attributes and safety features that are not sufficiently studied?

We have detailed, as an example, built environment attributes and safety features.

-For the problem or research gap/contribution of knowledge, it should be made clear. and the context or types of POS quality to be measured also needs to be clarified, e.g., urban parks or residential parks or both?

-Methodology is fine as the selected POSs are well justified. I am just thinking, the types and size of POS covered in the study are wide-ranging and diverse, how then this so called one size fits all assessment tool can be suitably and relevantly used? This needs to be discussed

Thanks for your comment. We have argued this into the discussion section.

-The statistical analysis is appropriate (EFA to validate the items under each factor) (unidimensional analysis). However, factor loading cut off needs to be clarified...whether the authors used 0.5 or 0.6 or lower?

We appreciate the opportunity to clarify. According to Field 2013, factor loadings less than 0.3 were suppressed. Scores greater than 0.4 were considered as stable (Guadagnoli and Velicer, 1988). We have added this information in the methods section and the relevant sources in references

-whether the alpha figures here are representing the reliability of each factor? What analysis has been used to ensure its reliability?

Thank you for your comment. Alphas show internal consistency of the latent variables. For this research the reference from University of Virginia. Using and Interpreting Cronbach’s Alpha was used to define the cut off values and it´s cited in the references.

For the results section, factor analysis and Cronbach alpha have been adopted but I noticed there is one particular factor with very low alpha value (less than 0.5, i.e., 0.15 for food and wellness environment), based on this, the items grouped under the factor are not consistency or reliable. I suggest removing them as this already indicated poor selections of items.

We decided to keep this particular factor in the manuscript. This is because a high amount of POS is surrounded by high dense food and/or beverages. We have deeply explained this in a paragraph within the discussion section.

-Since the authors also mentioned the importance of kappa value (measuring inter-rater reliability), why did not the authors carry out this with the 3 evaluators for the parks? Inter rater reliability is one of the key components ensuring the reliability and hence validity of the tool.

Thanks for the comment. This is one of two articles we are preparing for the validation of the MexPOS. We did first perform construct validity, once the instrument was restructured according to the results of the factor analysis, we will be using it to estimate the inter-rater reliability, reproducibility and face validity.

-Discussion needs to specifically highlight the similarities and differences that this study has contributed to the existing literature. The current piece is not clear.

Thanks! We have highlighted similarities and differences with other studies within the discussion section.

-Structure of sections is necessary that limitations and suggestions should be incorporated under the conclusion section.

We have incorporated strengths and limitations within the conclusion section.

Reviewer 4 Report

Please, find attached some suggestions in the file.

Author Response

Suggestions: I think that the research makes a good attempt in finding a model applicable to LMIC for evaluation of POS, starting from well-known and prestigious models applied to HIC. The fieldwork is extensive, and the statistical procedure is consistent. However, I think that the authors should make an effort to explain in more detail some of the steps in the procedure. In my view, the nature of the study makes it imperative to use graphical sources such as flow charts, maps, tables, etc., which allow for better monitoring and make it possible to reproduce the study in other similar contexts.

Here I make some suggestions that I think would enhance the quality of the paper:

About the typologies of POS.

1) metropolitan

2) local park: 3,000 to 10,000 m2

3) neighborhood park: 400 to 3,000 m2

4) pocket park: 100 to 400 m2

5) roundabout: has a higher proportion of sealed soil, vegetation or planters. Planned to regulate vehicular traffic

6) alameda: public space of up to 80,000 m2, with vegetation and bare or covered soil and paths or corridors for pedestrian traffic

7) plaza: public space of up to 5,000 m2, which has a greater proportion of sealed soil, has arboreal and/or shrubby vegetation. Planned for recreation, rest or relaxation

8) remnant: linear green space generally enabled along disused railway lines, rivers, streams, canals and urban voids, and

9) garden: public space of up to 5000 m2, which has a higher proportion of sealed soil, has arboreal and/or shrubby vegetation. Planned for recreation, rest or relaxation.

It would be nice to know how many POS have been analyzed from every typology. Would it be possible to find more detailed numbers?

Thanks for the valuable comment. We have included further information related to the typology within the results section. I am not sure that roundabouts or pocket parks could be comparable to metropolitan parks or gardens of 5.000 m2. For example, how likely is to find fountains or benches in the smallest ones or are safety and maintenance issues comparable? A paragraph has been included in the discussion section related to this information.

Maybe a map could help identifying the typologies (maybe with key of colours) or maybe a table with location and typology would be helpful.

In addition, we included a map of Mexico City with the distribution of typologies.

Some minor questions

Please, add the full name for the acronyms HIC and LMIC, the first time you include them in the text.

Thanks, we have added the full name for those acronyms in the document.

About the design of the model

I suggest adding a flow chart with the procedure, where the selection of the indicators appears clearly. Something like the following table or any other figure helping to understand the final model.

We have used your format to clearly indicate the procedure of the final model. Thanks!

The experts technique should be also better explained: how many experts, the expertise field, and years of expertise of each of them and which was the technique applied, maybe a simple brainstorming or whatever it was applied in every stage. Also, when the meetings were held. An example could be the following table, it could be also done by a Figure or any other graphical aid that the authors consider proper for clear explanation. I had to build it myself to understand the procedure, and this should be supplied to the reader. Statistical results could be added somehow to the chart. I have highlighted in red colour some aspects that should be clarified.

We have completed the figure you kindly proposed.

The tables for every model and the survey of model IV can be maintained as supplementary material, but a figure summarizing the whole process would be very helpful for the reader.

Thanks, we maintained appendixes and the supplementary material.

The final model, is it the one of the survey? According to the text, lines 120-126: The final version of the tool had the following sections: 1) General park information, 2) General park features (e.g. activities within the park, information & signalling, aesthetics), 3) Accessibility, 4) Environment around the park, 5) Roads/internal routes/tracks within the park, 6) Security, safety and lighting, 7) Facilities and amenities (e.g. toilets, drinking fountains, litter bins, benches), 8) Health, nutrition, and hygiene (e.g. medical services inside the park, food advertising, food establishments, hygiene-related to pets), and 9) Maintenance, does not fit with the factors in the survey, please explain it better.

We have changed “the final version of the tool” to “the original version of the tool” since this is the tool we used to evaluate the 944 POS.

Conclusion:

They are too concise. Maybe you could suggest other research to develop in the future having made this one. Something else, about the fieldwork that could be improved, maybe any other techniques that could be implemented after the expertise acquired in this research…

We have included further information in the conclusion section.

Maybe how to implement the results to propose an urban intervention in a specific area. The items examined could be used as indicators to monitor and track potential interventions… Have been the model applied to a case study validation? Please comment.

In addition to the previous comment, we have included some conclusions related to the potential use of this instrument.

Some ideas: to compare results among groups of population (children, women, the elderly…) There is no information about who should be the respondents to the survey. Maybe this could be future research, to reach inclusive POS, where everyone necessities are considered.

Thanks for this comment. However, the MexPOS is an audit tool and not a survey. So, we cannot compare population groups.

Round 2

Reviewer 2 Report

I don't have any further comments on the revised paper.

Author Response

Thanks for your comments.

Reviewer 3 Report

Dear authors,

Thank you for the revised manuscript. Significant improvement has been made; however, kindly revise the conclusion by adding in key salient findings obtained from this study and also kindly revisit the list of references as some of them are not well provided based on the format.

Author Response

We have revised and modified the conclusion section as well as the references.

Reviewer 4 Report

The manuscript has been improved as suggested. Thank you for considering the suggestions.

Author Response

Thanks for your comments.